# Nonuniform Correction of Ground-Based Optical Telescope Image Based on Conditional Generative Adversarial Network

**DOI:** 10.3390/s23031086

**Published:** 2023-01-17

**Authors:** Xiangji Guo, Tao Chen, Junchi Liu, Yuan Liu, Qichang An, Chunfeng Jiang

**Affiliations:** 1Changchun Institute of Optics, Fine Mechanics and Physics, Chinese Academy of Sciences, Changchun 130033, China; 2College of Optoelectronics, University of Chinese Academy of Sciences (UCAS), Beijing 100049, China

**Keywords:** deep learning, nonuniform correction, conditional generative adversarial network, optical telescope image

## Abstract

Ground-based telescopes are often affected by vignetting, stray light and detector nonuniformity when acquiring space images. This paper presents a space image nonuniform correction method using the conditional generative adversarial network (*CGAN*). Firstly, we create a dataset for training by introducing the physical vignetting model and by designing the simulation polynomial to realize the nonuniform background. Secondly, we develop a robust conditional generative adversarial network (*CGAN*) for learning the nonuniform background, in which we improve the network structure of the generator. The experimental results include a simulated dataset and authentic space images. The proposed method can effectively remove the nonuniform background of space images, achieve the Mean Square Error (*MSE*) of 4.56 in the simulation dataset, and improve the target’s signal-to-noise ratio (*SNR*) by 43.87% in the real image correction.

## 1. Introduction

Acquiring images using ground-based telescopes is of great significance to space situations. However, various optical phenomena often affect the imaging of detectors, resulting in image nonuniformity. The imaging nonuniformity of ground-based large-aperture telescopes is mainly caused by vignetting, stray light [1,2], and detector nonuniformity. Image degradation not only affects the image quality and image signal-to-noise ratio (SNR) but also has a severe impact on subsequent image segmentation and target detection. Therefore, optical system analysis and nonuniform background correction of space images are necessary pretreatment steps.

Vignetting is a common problem in optical systems, including natural vignetting, mechanical vignetting, and other effects [2]. Vignetting effects show that the energy captured by the detector decreases from the center to the edge of the image. Stray light is another influencing factor in telescope imaging. It refers to the background radiation noise generated by the light rays in the non-imaging field of view region of the system image plane. Stray light reduces the signal-to-noise ratio (SNR) of the target, thus affecting the detection or recognition ability of the whole system, or ensuring the detected target signal is wholly eradicated in the background of stray light. 

Much research has been done on the problem of vignetting correction in imaging systems. The simplest method is flat field correction with uniform illumination [3,4,5]. Although flat field correction can solve the vignetting problem in a fixed scene, it must be corrected again when it changes. It is relatively tricky for large aperture telescopes to obtain flat field images with uniform illumination. Another method to resolve the vignetting is to process the image. Methods can be divided into multi-frame image methods or single-frame image methods according to the number of image processing. The traditional multi-frame image method extracts vignetting information from multiple images in the same scene. Yuan et al. [6] proposed an improved radial gradient correction method, and the corrected images have better radial uniformity and more precise details. Goldman et al. [7] discussed whether the response curve was known and used polynomial fitting to remove vignetting in multi-frame images. Litvinov et al. [8] proposed a method for simultaneously estimating camera radiation response, camera gain, and vignetting using multiple image frames.

Single-frame vignetting correction methods are more flexible because they do not require multi-frame information accumulation. Zheng et al. [9] realized vignetting correction by the segmentation method, but the correction effect of this method is affected by segmentation accuracy. The method based on the radial brightness channel proposed by Cho et al. [10] has a faster processing speed, but the application of problem scenarios is limited. Lopez-Fuentes et al. [11] corrected the vignetting distortion by minimizing the logarithmic intensity entropy of the image, but the problem of overcorrection often existed. The influence of stray light on imaging is random. Generally, researchers can reduce the impact of external and internal stray light by setting an optical structure to suppress stray light [12].

The methods based on pre-calibration or iteration have problems such as being only suitable for a single scene or a large amount of calculation. In recent years, with the wide application of deep learning in various fields, optical problems are also more widely solved by deep learning, including optical interferometry [13], single-pixel imaging [14,15], wavefront sensing [16,17,18], remote sensing [19,20,21,22,23,24,25,26] and Fourier ptychography [27,28,29]. Using deep learning for image enhancement in imaging systems is also more attractive [22,23,30,31,32]. Chang et al. [33] applied the deep residual network to infrared images and showed good robustness to vignetting and noise-induced nonuniformity. Fang et al. [34] applied U-Net to nonuniform correction to correct the imaging shortcomings of detectors by learning ring artifacts. Jian et al. [35] introduced the filter into the convolutional neural network to optimize the image for the high-frequency inhomogeneity in the infrared image. Zhang et al. [36] proposed a Shearlet deep neural network and defined regularization methods based on shearlet-extracted features to help image detail restoration and noise removal.

Generative adversarial networks are widely used in image generation and enhancement due to their excellent fitting performance. Zhang et al. [37] applied a conditional generative network to the problem of rain removal, which can effectively improve the accuracy of target detection in subsequent rain. Armanious et al. [38] have proposed MedGAN, a new medical image-image conversion framework, which is superior to other methods for different image conversion tasks. Dai et al. [39] proposed an adversarial network based on residual cycle generation to correct the nonuniformity of medical images. Various correction parameters have been improved to some extent compared with traditional GAN and U-Net. Kuang et al. [40] used generative adversarial networks to correct the nonuniformity caused by optical noise in infrared images and achieved excellent results in various infrared images.

This paper proposes a correction algorithm based on a conditional generative adversarial network (*CGAN*) to solve the imaging nonuniformity problem of ground-based telescopes. Since no dataset is available for training, we create a data set containing nonuniform image pairs. Firstly, according to Kang-Weiss’s [41] vignetting model, we randomly add vignetting to the image by changing the hyperparameter space. At the same time, Zernike polynomial and sigmoid functions are introduced to simulate the image background nonuniformity caused by stray light or detector nonuniformity. Furthermore, we modify the *CGAN* to extract background information better, applying a cascade network and extremely efficient spatial pyramid (EESP) to improve the performance of the generative network. The nonuniform background is inferenced by a trained network, and the pure space image is obtained by removing the background. 

The arrangement of this paper is as follows. Section 1 introduces the correlative methods and research significance of vignetting and nonuniform correction. Section 2 introduces the basic process of nonuniform correction and the principle of GAN. We present the proposed method in detail in Section 3, including the design of the network structure and the simulation method of the nonuniform background. Section 4 shows the test results of our approach on simulation datasets and discusses various comparisons. Meanwhile, we test the algorithm with real images and make a comparative analysis with multiple methods. In Section 5, we discuss the simulation and our experimental results. Finally, we summarize the work and contribution of this paper in Section 6.

## 2. Preliminaries

An imaging system with interference from an optical system can be expressed as:
(1)$$I^{\prime }=I+V\cdot S+n=I+B$$ where I and $I^{\prime }$ are the pure image and the image with the nonuniform background. V is vignetting background, S is the nonuniform background caused by stray light or detector, and *n* is the detector noise. Assuming that the vignetting changes slowly and evenly and stray light will not cause the abnormal exposure of some images, So the corrected image can be obtained by this formula:(2)$$I=I^{\prime }-B$$ where B is the nonuniform background. We aim to train a robust model to learn information about a nonuniform background from a large number of images:(3)$$B=F(I^{\prime })$$ where F is the network model used for inferencing nonuniform background after supervised learning. The nonuniform image is the non-exposure distortion image. In other words, there is no significant number of saturated pixels in the image, only the maximum pixels at the corresponding points of space targets or stars.

Generative adversarial network (GAN) is a generation algorithm based on the deep learning model proposed by Goodfellow et al. in 2014, which can be used for various generation tasks. GAN includes two networks, a generator (*G*) and a discriminator (*D*). In the field of image generation, the generation network is used to generate pictures, and its input is a random noise (*z*), through which the image is generated, denoted as *G* (*z*). The discriminator is used to determine whether an image is real or not.

In the training process of GAN, it is hoped that the images generated by the generated network *G* can be as real as possible and can deceive the discriminant network *D*. The discriminant network *D* is expected to distinguish the images generated by *G* from the real ones. For the GAN framework, its value function V(G,D) is: (4)$$\underset{G}{\min}\underset{D}{\max}V(D,G)=E_{y\sim p_{data}(y)}[\log D(y)]+E_{z\sim p_{z}(z)}[\log(1-D(G(z)))]$$
where y is the real image, The trained generation network *G* can therefore be used to generate “fake” pictures.

## 3. Materials and Methods

### 3.1. Network Structure

Inspired by Isola et al. and Armanious et al. [38,42], we introduce conditional generative adversarial networks (*CGAN*) to solve the problem of space image nonuniformity. As shown in Figure 1, the network consists of residual-casnet (generator) and patchgan (discriminator). Unlike other end-to-end generative adversarial networks that directly generate corrected images, the nonuniform background is developed for two main reasons. Firstly, the background is gentler than the space image, so it is easier for the network to learn the background features. Secondly, the nonuniform background is assumed to be smooth. Images with alterable resolution can be sent into the background inference to adapt to different application scenarios.

The generator contains four U-blocks, and the skip connection between different U-block realizes the information fusion. The image resolution is 2048 × 2048, and it is challenging to train due to memory overflow when directly sent to the generator for training. The original image sent into the network is compressed to 1/2 of its original size. Average pooling reduces the input resolution, and the network’s output needs to be further fed into the discriminator network. In this case, bilinear interpolation is used to restore the original resolution.

Similar to the structure of the U-Net [43], U-block is a fully convolutional encoder and decoder structure. In the encoding part, we introduce the extremely efficient spatial pyramid [33,44] (EESP) to increase the receptive field with the minimum number of parameters and improve the performance of the network. Batch Normalization (BN) and max-pooling are applied to avoid overfitting and to achieve downsampling after the EESP structure. The decoding section uses deconvolution with stride 2 to implement upsampling.

The discriminator in traditional GAN can judge the input’s probability as either a generated fake or a real image. Although the discriminator improves network performance in competition with the generator, the discriminator does not specify which part of the image affects the probability of output. The output of our discriminator is the average value of N × N patches like patchgan proposed in the literature [42]. Specifically, the discriminator structure is composed of 5 convolution layers, the stride of the first three layers is 2, the stride of the last two layers is 1, and finally the discriminator generates a tensor with a size of 128 × 128 × 1. The discriminator improves the inference effect of the generator in more detail by judging every point in the patch.

### 3.2. Training Strategy

Adam is used as the optimizer for network training. The value of the exponential decay rate estimated by the first moment is set as 0.5, and the initial learning rate is set as 0.0002. Considering the resolution of network input, we set the number of batches of network training as 2, with a total of 200 rounds. 

The generator and discriminator are trained in cycles to obtain better network performance. To train the generator better, we propose a Joint loss function. The loss functions are shown below:(5)$$G^{*}=\arg\underset{G}{\min}\underset{D}{\max}L_{CGAN}(G,D)+\lambda L_{L1}(G)+\gamma L_{SSIM}(G)$$
where $L_{L1}$ is MAE loss, which is used to evaluate the regression of corresponding points after network mapping. $L_{CGAN}$ is the loss generated for the discriminator. $L_{SSIM}$ represents the loss of structural similarity (*SSIM*) between input and output, and can be combined with the results of the discriminator to make the generator more accurate in detail and produce images that are more similar to the real background. λ and γ are parameters to adjust the proportion of the two loss functions. λ and γ are set to 10 and 5 for better network performance in this paper. The three kinds of losses are expressed as follows:(6)$$L_{L1}(G)=E_{x,y,z}[{\left\|y-G(x,z)\right\|}_{1}]$$
(7)$$L_{CGAN}(G,D)=E_{x,y}[\log D(x,y)]+E_{x,z}[\log(1-D(x,G(x,z)))]$$
(8)$$L_{SSIM}(G)=1-SSIM$$
(9)$$SSIM=\frac{\left(2\mu_{x}\mu_{y}+c_{1})(2\sigma_{xy}+c_{2}\right)}{\left(\mu_{x}^{2}+\mu_{{}_{y}}^{2}+c_{1})(\sigma_{x}^{2}+\sigma_{{}_{y}}^{2}+c_{2}\right)}$$
where $\mu_{x}$ and $\mu_{y}$ represent the average of the input (*x*) and the output (*y*), $\sigma_{x}$ and $\sigma_{y}$ represent the standard deviation of *x* and *y*, and $\sigma_{xy}$ is the covariance of x and y. $c_{1}$ and $c_{2}$ is a constant that avoids errors when the denominator is 0. D(x,y) represents the loss of the discriminator, and the cross-entropy is used as the loss function in this paper.

### 3.3. Nonuniform Models and Datasets

Training of the network needs image pairs from ground-based telescopes, but pairs of data are often hard to get, so it is necessary to construct images for training and validation. Selecting space images (I) not affected by nonuniform influence can be used for training by adding nonuniform impact (φ). Considering the nonuniformity caused by optical phenomena, the nonuniform image $\left(I^{\prime }\right)$ can be obtained as follows: (10)$$\varphi =\eta V\cdot S+n,I^{\prime }=I+\varphi$$
where V and S represent vignetting and nonuniformity caused by stray light or detector. n is random noise. The noise added in this paper is mainly Gaussian noise and Poisson noise of different degrees. η is the hyperparameter regulating the degree of nonuniformity. 

The physical model can realize the effect of vignetting. For the distance from the center of vignetting $r={\left(u^{2}+v^{2}\right)}^{1/2}$, u and v are the horizontal and vertical distances. According to Kang-Weiss’s model, the vignetting model is shown in the following formula:(11)V=A⋅G⋅T
where A is the background matrix caused by off-axis illumination attenuation, G is the vignetting matrix caused by the optical inherent error of lens group, and T is the corresponding matrix of tilting the camera, The expressions of A, G and T are as follows:(12)$$A=\frac{1}{{\left(1+{\left(r/f\right)}^{2}\right)}^{2}}$$
(13)G=1−αr
(14)$$T=\cos\tau {\left(1+\frac{\tan\tau }{f}(u\sin\chi -v\cos\chi )\right)}^{3}$$
where f and α are the effective focal length and the parameters describing lens-induced vignetting. τ and χ are the angles at which the camera is tilted at two angles. 

Zernike polynomials consist of an infinite number of complete sets of polynomials, often used to describe wavefront properties. Similarly, the superposition of multiple Zernike polynomials can simulate the random fluctuation of a two-dimensional surface. Further, we construct a polynomial with the sigmoid function, Equation (15) shows the function value caused by simulated stray light:(15)$$S=[\mu \sum_{j=1}^{n}\lambda_{j}Z_{j}(u,v)+\frac{1}{1+\exp(-\sum_{j=1}^{n}\lambda_{j}Z_{j}(u,v))}{]}_{nor}$$
where Z(u,v) represents different Zernike polynomials, λ is the corresponding coefficient. We choose the first 30 Zernike polynomials, i.e., n equals 30. μ is the parameters to adjust of two polynomials. A variety of nonuniform gradient backgrounds can be obtained by changing the value of μ. As shown in Figure 2, the nonuniform background contour is precise when μ is small, and when μ increases gradually, the nonuniform background is also steadily blurred.

The polynomial can better simulate the nonuniform background under different conditions, and *nor* represents the normalization of the parameter matrix. The common Zernike expression is:(16)$$\begin{array}{l} Z_{1}=v,Z_{2}=u,Z_{3}=2uv\\ Z_{4}=2(u^{2}+v^{2})-1,Z_{5}=u^{2}-v^{2}\\ \ldots \ldots \end{array}$$

The image pairs used for training are represented in Figure 2. Different nonuniform backgrounds can be obtained by changing the parameter space $\xi =\left\{f^{-1},\alpha ,\tau ,\chi ,\mu ,\eta \right\}$. For the training of the network, the generated image pairs are selected as the dataset, among which 50,000 pairs of images are set as the training set to train the convolutional neural network, 10,000 pairs of images are used as the validation set, and we use 1000 images as the test set. 

## 4. Results

The experiment of the proposed method consists of two parts: the test of a synthetic nonuniform dataset and the test of real nonuniform images. We first test the simulation dataset and analyze the performance of nonuniform correction under various conditions. Then the real image is tested and compared with the existing nonuniform correction algorithms. Our equipment for training and testing data is a high-performance computer with AMD 3600X CPU and Nvidia GTX2080ti GPU.

### 4.1. Evaluation Metric

There is ground truth in the simulation data but not in the real test data, so the evaluation index is also different in the two experiments. Structural similarity (*SSIM*) and mean square error (*MSE*) are used to evaluate the correction performance of the simulated nonuniform dataset. The structural similarity is shown in Equation (9). Furthermore, the mean square error is defined as follows:(17)$$MSE=\frac{1}{m\times n}\sum_{i=0}^{m-1}\sum_{j=0}^{n-1}{\left(I(i,j)-C(i,j)\right)}^{2}$$
where *m* and *n* are the resolution of the image, respectively, which are both 2048 in this research. *I* and *K* are the true uniform image and the corrected image. MSE represents the difference between two images, so the smaller the value, the better the correction effect; *SSIM* represents the similarity of two images, and the larger the value, the better the correction effect.

There is no corresponding ground truth in the real space nonuniform image. We cannot use the above two evaluation metrics. Therefore, we introduce two other evaluation metrics to verify the correction effect of real images. The residual error of the corrected image is used as one of the evaluation criteria. First, we use the adaptive threshold segmentation algorithm to accurately obtain the threshold *T* of the corrected image:(18)$$T=\mu_{cr}+\alpha \sigma_{cr}$$
where $\mu_{cr}$ and $\sigma_{cr}$ are the mean and standard deviation of the corrected image, respectively, α is the hyperparameter for adjusting the threshold, and the value of a is 1 in order to reduce the interference as much as possible. The residual image is obtained as follows.
(19)$$R(i,j)=\left\{ \begin{array}{l} 0,C(i,j)\geq T\\ C(i,j),C(i,j)<T \end{array}\right.$$

Finally, we choose the mean $\left(\mu_{re}\right)$ and standard deviation $\left(\sigma_{re}\right)$ of the residual image as the evaluation criteria. Using the residual image can effectively evaluate the overall situation of the correction. Furthermore, we introduce the target signal-to-noise ratio as an index to evaluate the local effect. The global standard deviation and signal-to-noise ratio (*SNR*) were used to evaluate the quality of the corrected image. Signal to noise ratio is defined as follows:(20)$$SNR=\frac{\mu_{r}-\mu_{B}}{\sigma_{B}}$$
where $\mu_{r}$ is the mean value of the target region, $\mu_{B}$ is the mean value of the background region, and $\sigma_{B}$ is the standard deviation of the background region. The performance of the algorithm can be effectively evaluated by the residual error of the corrected image and the signal-to-noise ratio (*SNR*) of the target in the local area.

### 4.2. Result on Test Set 

The test data in this part is the dataset described in 3.3, which contains 1000 images with different nonuniform backgrounds. The correction effect of the proposed method is shown in Figure 3. It can be seen from the Figure that most of the nonuniform background in the image is removed, and an image with a relatively pure background is obtained. At the same time, the high-frequency components of the image are well preserved, which is conducive to the subsequent processing of threshold segmentation and object detection. Since the network generates the background, we can intuitively see that the energy distribution of the fitted background and the real background is almost the same by obtaining the heat map of the nonuniform background, i.e., most of the gray values at different positions are the same or have little difference. In quantitative analysis, we get good results on this test set, with *MSE* of 4.56 and *SSIM* of 0.99, respectively.

### 4.3. Ablation Experiments

In this subsection, we conduct two ablation experiments to verify the effectiveness of the network structure and the influence of different loss functions on the results, respectively.

#### 4.3.1. Ablation Experiment on Generator

Compared with the traditional pix2pix [42] network, the proposed method mainly makes improvements to the generator structure. To explore the influence of different structures on network performance, we train generators with different structures and test them on the same dataset. The generators of different structures are: (a) simple codec structure in pix2pix (SCSP); (b) U-net structure (US); (c) casnet with simple concatenation of four U-blocks (CSCU); (d) casnet adding the same skip connection as in this paper (CASC); (e) adding EESP structure on the basis of Unet (AESU); (f) structure of the generator in this paper (Ours).

The experimental results are shown in Table 1, and different generator structures possess different performances. Compared with the simple codec structure, Unet has a more obvious improvement, which is mainly due to the skip connection operation, and the use of shallow features improves the performance of the network. Casnet does not show a significant improvement on Unet, but the corrected results with the same addition of skip connection structure show a specific improvement. At the same time, we see in the experiment of group e that the EESP structure is beneficial to U-block, and the enlarged receptive field makes the network pay more attention to the information of the context. Finally, our generator achieves the best results in the efficient module which combines the cascade structure, skip connection structure and EESP.

#### 4.3.2. Ablation Experiment on Loss

We conduct ablation experiments to explore the influence of different loss functions on the results. The experimental results are shown in Table 2. Three loss functions have different emphases in the process of backward gradient propagation. When L1 is used as the loss function, the loss function is realized by calculating the deviation of all points between the output and the ground truth. As one of the standard loss functions in regression tasks, the L1 loss can effectively evaluate pixel-level regression. When *SSIM* is used as the loss function, the network will pay more attention to the overall regression effect of the data because the mean and variance of the data are used in the calculation. However, when the generator only uses the loss function of the discriminator, the network will pay attention to the regression effect in each patch. All three loss functions can effectively train the network to a certain extent, but the L1 loss function has the best impact in a single training. The combined loss effect of L1 and *SSIM* loss is the best when the two loss functions are fused, but the best correction effect is also improved after *CGAN* loss is added.

To further determine the proportion of Joint loss function, we have carried out the other experiment. We found that the loss function that contributed the most to the results was the L1 loss, so this experiment focuses on exploring the best ratio of L1 loss to the other two loss functions. Table 3 demonstrates the correction results for different loss function proportions. Through experiments, we find that λ and γ in formula (5) can obtain the best results when 10 and 5 are set respectively, and the two evaluation indicators reach the best at this time. Due to the limited number of experiments we have undertaken, it is possible that the ratio of the loss function can be further improved with further fine-tuning. Still, small changes do not have much impact on the results, and this experiment is only about order(s) of magnitude changes.

### 4.4. Analysis of Patch Size

This section experiments on the influence of patch size on the discriminator’s result. As mentioned in the literature [42], different patch sizes will impact the final result. Convolution with a step size of 2 is used to change the feature map size instead of pooling operation in the discriminator. The final patch size is altered by changing the number of convolution layers.

Figure 4 shows the result corrected by the network trained with different patch sizes for the same nonuniform image. Most of the nonuniformity in all images is removed, which benefits from the loss function of L1 loss and *SSIM* loss. However, there are still differences in the corrected images. As the patch size increases from the minimum, the network’s correction results also gradually improve. However, the effect of correction is the best when the patch size is 128 × 128. Subsequently, the correction results are slightly reduced with the increase in patch size. Although there is little difference in the overall image correction effect, the increase in patch size will also cause longer training time, so 128 × 128 is finally selected as the output of the discriminator.

### 4.5. Impact of Resolution

The generator’s network does not contain a fully connected layer. Multiple resolutions of input can be accepted. Therefore, in this section, we discuss the effect of learning different resolutions on the nonuniform correction results, where the input of the discriminator is fixed, and the input resolution of the generator is changed. We use max pooling and bilinear interpolation to adjust the input and output resolution size. The nonuniform correction effects of several different resolutions are shown in Table 4. With the increase of the image size of the input generator, the image correction effect is significantly improved. Still, the original size does not improve much compared with the half resolution. 

Due to the test set containing different types of nonuniform images and different gradient nonuniform backgrounds, the sensitivity of the samples is also different. The sensitivity of nonuniform background with varying gradients to pooling is not precisely the same, and the number of pooling in the image with slow background change will not influence the final result. On the contrary, image pooling with a fast-changing background will lose a lot of background information. In practical application, it is possible to select appropriate subsampling times according to the background gradient of the image to realize the correction of different images, which is a significant advantage of network learning background instead of learning uniform images directly.

### 4.6. Experiment on Real Images

The images we test in this part are the space images taken at night by the ground-based telescope, whose specific parameters are shown in Table 5. We select images taken at different times and under different skylight conditions as tests, a total of 100 images. Images all contain space targets to ensure that the target SNR can be used as an evaluation criterion. We compare two traditional methods and two recent deep-learning nonuniform removal methods. The traditional methods are the Top-Hat transformation (THT) method [45] and the mean iterative filtering (MIF) method [46]. And the method of Kuang [40] and Dai [39] are chosen as deep learning methods. Among them, Kuang is also a method for generating adversarial networks, and the main difference from the method in this paper is the network structure. A quarter of the resolution is used for training because of its large memory footprint in method Dai.

The comparison is shown in Figure 5, in which images (I) and (II) have a gentle nonuniform background and have little influence. Several methods can remove the nonuniform background in the image to a certain extent. The THT method cannot remove the residual image in the corrected image. The image (I) still has some nonuniformity after correction by this method, and the upper right part of the image (II) has an over-correction phenomenon. Although MIF can remove most of the nonuniform background well, the over-aggressive correction reduces the gray value of stars and targets, which is undesirable. Methods based on deep learning can achieve an almost perfect correction effect. However, there are apparent differences between images (III) and (IV) after correction. Two traditional methods still have the above problems. Meanwhile, the images of the two deep learning methods will still have some nonuniform residual shadows after correction, and our approach has achieved relatively ideal correction results.

Furthermore, we enlarge the correction results of the image (IV) by different methods in Figure 6. We select a local region with space targets in the red box containing three space point targets. The SNR of the space target is high, but the detection and tracking of the subsequent point target are difficult due to the sizeable nonuniform background of the overall image. THT reduces the gray value of space targets and stars while eliminating the nonuniform background, reducing the target SNR. The effect of the MIF method is similar to THT. The gray value is reduced, but the image corrected by the MIF method is less noisy. Kuang can effectively remove some nonuniform image backgrounds while preserving image details. Dai effectively drew most of the nonuniform background. At the same time, introducing more noise will also bring trouble to the subsequent processing. Our method is further improved based on the Kuang method. Not only the corrected image is more uniform on the whole, but also the details at the high frequency of the image are more distinct.

We quantitatively analyze the correction results of various methods. Specific data are shown in Table 6. $\mu_{re}$ and $\sigma_{re}$ are the mean and variance of the residual image described in Section 4.1. Table 6 shows the specific correction results of different methods, and its specific data is consistent with the correction effect in Figure 6. We can see that the values of $\mu_{re}$ and $\sigma_{re}$ are equally small for images with less nonuniformity and less noise after correction. MIF also achieves good results, but this is at the cost of reducing the target gray value. The deep learning-based method has better retention of targets and stars because it can make the network autonomously learn image inhomogeneity.

Although the image residual can describe the correction effect of the image to a certain extent, it is not necessarily the case that the smaller the value is, the better the correction effect. When the algorithm corrects all the pixels of an image to 0, the values of $\mu_{re}$ and $\sigma_{re}$ will be minimized, which is not what we want. Therefore, we introduce the target SNR to verify the effect of image correction further. One space target is selected from each image as the detection sample. Finally, the SNR of all targets is averaged to obtain the target SNR after correction by different methods. The target SNR can fully characterize the local effect after image correction, and our method achieves the highest result, consistent with Figure 5 and Figure 6. 

## 5. Discussion

Simulation and experimental results show the effectiveness of the proposed method. Firstly, we conduct ablation experiments to verify the proposed network structure’s effectiveness and the loss function’s rationality. The proposed network structure combines the cascade method and the idea of skip connection to effectively improve the performance of the network. At the same time, compared with the loss function used in traditional *CGAN*, we further discuss the possibility of the loss function. *SSIM* can evaluate the degree of training, thus improving the training accuracy. Furthermore, we analyze the influence of different patch size in the discriminator, and the results are similar to many *CGAN* algorithms. Finally, we verify the applicability of the algorithm by feeding images with different resolutions into the generator. We find that the sensitivity of different nonuniform backgrounds to different resolutions is also very different. The image with a gentle gradient of nonuniform background can still obtain good results through the network after downsampling. When the gradient of the background is large, only a tiny amount of downsampling is used to ensure the integrity of the nonuniform background.

We tested the network trained on the simulated dataset on real images and achieved very good results. This is mainly due to the huge amount of simulation dataset and the diversity of simulation data, and a variety of different nonuniform backgrounds greatly improve the generalization of the network. In our experiments, we found that when the diversity of the simulation data set is poor, it will cause the phenomenon of “overfitting”, that is, the performance is magnificent in the simulation dataset, but the correction effect is poor in the real image.

In the process of constructing the dataset, we also tried some ray tracing software for simulation. However, this method not only limited the nonuniform background simulation, but also often failed to effectively simulate the general situation by using limited optical systems, resulting in the limitation of the dataset. Space image nonuniform correction using supervised learning methods is extremely dataset dependent, so subsequent research may prefer unsupervised learning methods.

## 6. Conclusions

This paper proposes a space image nonuniform correction algorithm based on the conditional generative adversarial network. This paper systematically discusses the construction method of a nonuniform image dataset, the introduction of *CGAN*, the application procedure of using *CGAN* for nonuniform correction, and the validation of the effectiveness of the *CGAN* (simulations and experiments). Some conclusions are presented below:The proposed method achieves the ideal results in the simulated dataset. Furthermore, we discuss the impact of the loss function, patch size, and input image resolution on the results and prove the effectiveness and applicability of the proposed method.In the real image nonuniform correction, the network trained by the simulation dataset can effectively remove different nonuniform backgrounds. Compared with the current deep learning-based nonuniform correction methods, our method achieves the best visual effects and quantitative analysis results.

The method is suitable for applications where the nonuniform background changes with the telescope pointing or the telescope is fast-tracked. This work will contribute to applying deep learning to space image nonuniform correction.

## Figures and Tables

**Figure 1 sensors-23-01086-f001:**
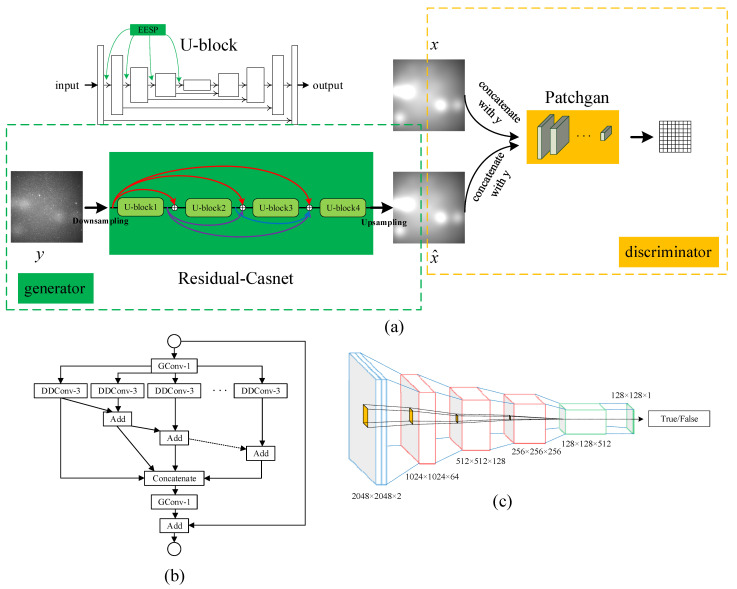
The overall structure of the network. (**a**) the design of the *CGAN*, including the generator and discriminator, U-block is the basic unit of the generator; (**b**) the EESP structure in U-block, and the receptive field can be increased with the smaller parameter; (**c**) the structure of patchgan.

**Figure 2 sensors-23-01086-f002:**
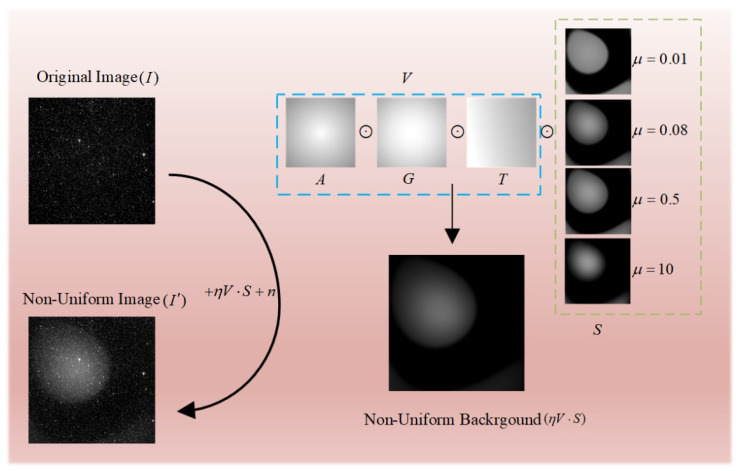
Schematic diagram of nonuniform image generation. The Figure shows different nonuniform backgrounds generated by taking various parameters of μ, and the final image is the background when μ equals to 0.08.

**Figure 3 sensors-23-01086-f003:**
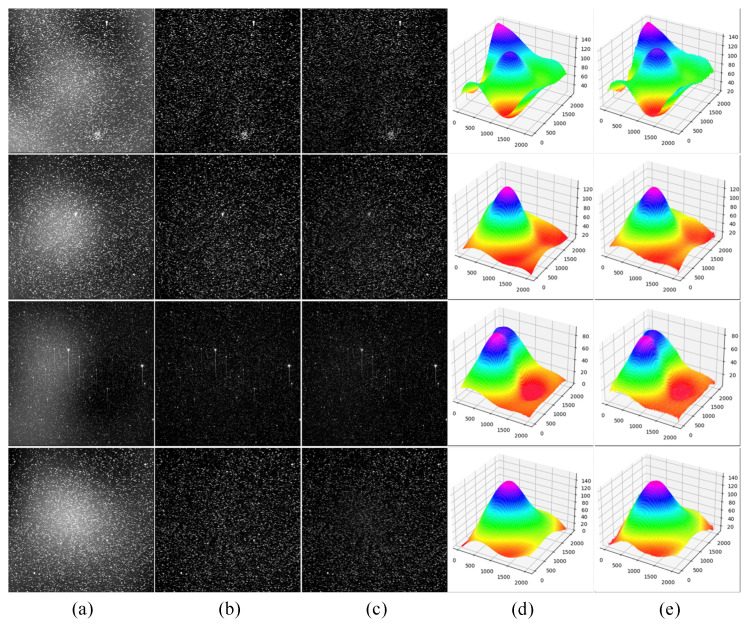
Nonuniform correction effect diagram. (**a**) represent generated nonuniform images; (**b**) represent uniform images; (**c**) represent corrected images; (**d**,**e**) are heat maps of nonuniform background ground truth and heat maps of inferenced background, respectively.

**Figure 4 sensors-23-01086-f004:**
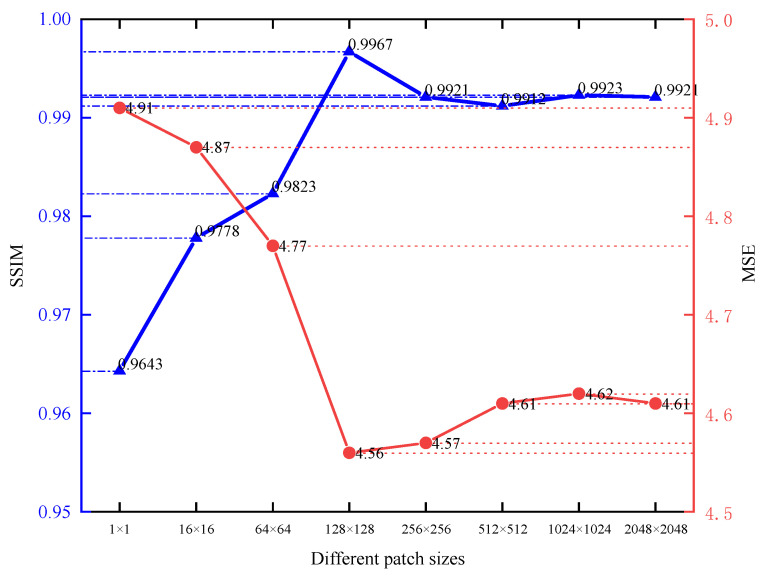
Nonuniform correction results obtained with different patch sizes.

**Figure 5 sensors-23-01086-f005:**
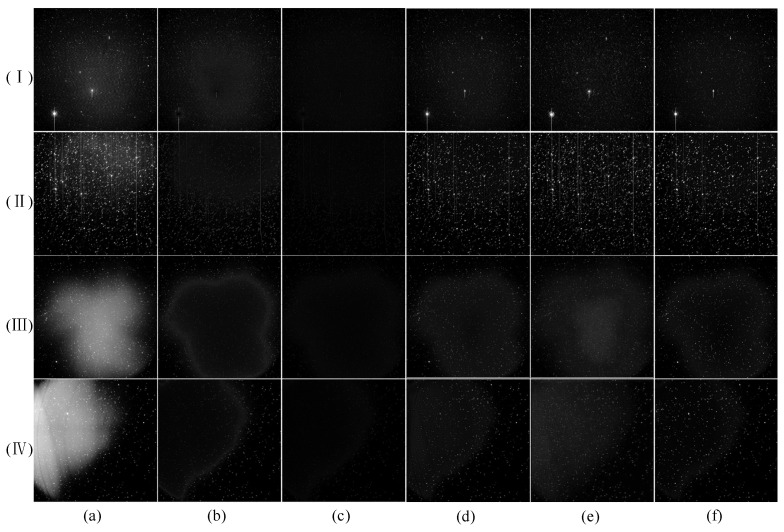
Comparison of correction effects of various methods for real nonuniform images. (**I**–**IV**) show different images. (**a**) is the original image, and (**b**–**f**) are the correction effects obtained by THT, MIF, Kuang, Dai and our method, respectively.

**Figure 6 sensors-23-01086-f006:**
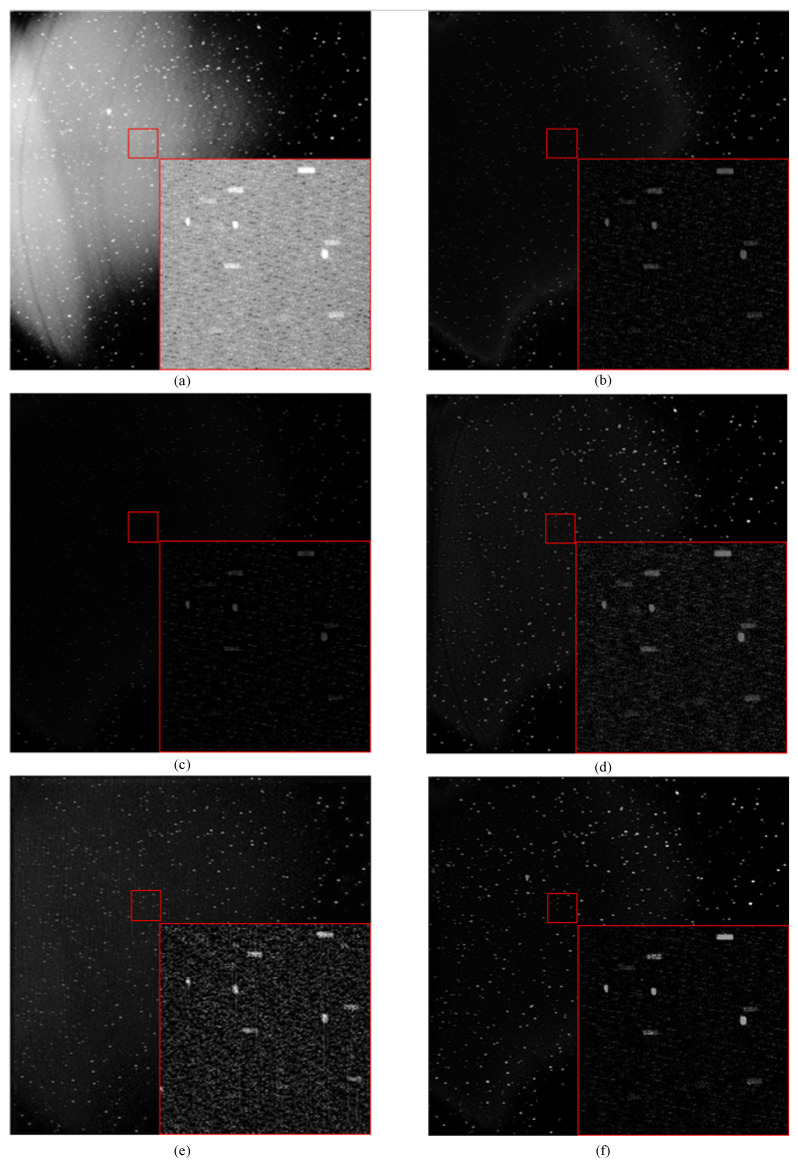
Detail images of different methods in (IV), (**a**–**f**) are the corresponding images of row 4 in Figure 5.

**Table 1 sensors-23-01086-t001:** Performance of the different generators in the test set.

	SCSP	US	CSCU	CASC	AESU	Ours
*MSE*	10.45	8.54	7.94	5.43	6.98	4.56
*SSIM*	0.92	0.91	0.92	0.98	0.95	0.99

**Table 2 sensors-23-01086-t002:** Ablation experiments of the three loss functions. All the loss functions are equally distributed when the loss functions are fused.

*CGAN* Loss	L1 Loss	*SSIM* Loss	*MSE*	*SSIM*
√			10.09	0.95
	√		4.93	0.98
		√	9.91	0.96
√	√		4.90	0.99
	√	√	4.88	0.97
√		√	9.61	0.96
√	√	√	4.66	0.99

**Table 3 sensors-23-01086-t003:** Influence of different proportional loss functions on the results, where the proportional coefficients represent L1 loss, *SSIM* loss and *CGAN* loss, respectively. Since the *SSIM* values of different groups differ very little, we show four significant digits.

	1:1:1	10:1:1	10:5:1	10:1:5	100:1:1	100:5:1	100:1:5
*MSE*	4.66	4.59	4.56	4.58	4.67	4.64	4.64
*SSIM*	0.9965	0.9965	0.9967	0.9964	0.9954	0.9955	0.9961

**Table 4 sensors-23-01086-t004:** Running time and correction results for different resolution inputs.

	Inference Time	*SSIM*	*MSE*
256	10.5 ms	0.9223	10.41
512	11.2 ms	0.9312	8.59
1024	12.4 ms	0.9967	4.56
2048	15.2 ms	0.9965	4.52

**Table 5 sensors-23-01086-t005:** Relevant parameters of the ground-based telescope used to acquire the image.

Parameter	Value
Focal length	48 m
Aperture	2 m
CCD Pixel size	13 μm × 13 μm
Total pixels	2048 × 2048
Central wavelength	550 nm

**Table 6 sensors-23-01086-t006:** Correction results of real nonuniform images by various methods.

	Original	THT	MIF	Kuang	Dai	Ours
$\sigma_{re}$	45.48	5.29	1.82	6.86	9.98	**0.92**
$\mu_{re}$	24.51	2.33	0.46	3.33	7.68	**0.26**
*SNR*	9.8	5.9	9.7	10.9	7.7	**14.1**

## Data Availability

Not applicable.

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
