# Peer review of "Nonuniform Correction of Ground-Based Optical Telescope Image Based on Conditional Generative Adversarial Network"

_sensors, 2023, doi:10.3390/s23031086_

Round 1
Reviewer 1 Report
As mentioned in astronomy, the paper is mainly about flat field correction.
The astronomical methods for flat-field correction is quite simple by imaging the sky near dusk or dawn, then we may subtract the dark current from each individual flat-field frames.
Hence the metrology described in the paper may be way too complex for flat-field correction.
However, it is a fine method for non-uniform background correction.
It is better for the author to change the description. For instance, the method may fit the application when the non-uniform background change versus telescope pointing or when telescope is fast tracking.
Reviewer 2 Report
In this paper, the authors proposed the conditional generative adversarial network for space image nonuniform correction. Firstly, a data set was created. Secondly, the author improves the network structure of generators in conditional generation confrontation network. The experimental results show that the proposed method can effectively remove the nonuniform background of space images. However, the following problem need to be improved.
(1) The English of paper is poor, please polish it again.
(2) All variables in the formula should be described.
(3) Please give all parameters of the proposed method.
(7) More related references should be cited, such as
[*]MRDDANet: A multiscale residual dense dual attention network for SAR image denoising[J]. IEEE Transactions on Geoscience and Remote Sensing, 2022, 60: 1-13.
(9) In Section 6, the font of the word "ideal" is not uniform on page 14 in line 495. Please adjust the font.
(10) Section 3 contains "3.1 Network structure", "3.3 Training strategy" and "3.4 Nonuniform models and datasets". Where is the content of section 3.2? Or is there a problem with the serial number?
(11) On page 6 , "Where A" in line 216, "Where f" in line 222 and "where Z(u,v)" in line 230. Please check the case of "Where" and the case of "where" in other places. The case of the English letter "W" is a bit confusing. Please check it, including letter case problems elsewhere.
(12) The layout of Figure 1 is a bit ugly. Can you adjust the layout of Figure 1?
(13) On page 5, please explain the abbreviation of "SSIM" in Formula (9).
(14) On page 7, "4. result" on line 250. Please adjust the case of the letter "r".
(15) It is recommended to replace "a b c d e f" in Table 1 with the abbreviation of network structure.
(16) A brief introduction to Table 3 should be given before Table 3, for example, Table 3 shows····
Round 2
Reviewer 2 Report
The respond is OK.